# Investigating the accuracy of Apple Watch VO$_2$ max measurements: A validation study

Rory Lambe[1,2*], Ben O'Grady[1,2], Maximus Baldwin[1,2,3], Cailbhe Doherty[1,2]

**1** School of Public Health, Physiotherapy and Sports Science, University College Dublin, Dublin 4, Ireland, **2** Insight Research Ireland Centre for Data Analytics, University College Dublin, Dublin 4, Ireland, **3** Institute for Sport and Health, University College Dublin, Dublin 4, Ireland

\* rory.lambe@ucdconnect.ie

## Abstract

VO$_2$ max is a measure of cardiorespiratory fitness and a key indicator of overall health. It is predictive of cardiovascular events and shows a strong inverse association with all-cause mortality. Increased cardiorespiratory fitness is associated with reductions in coronary artery disease, diabetes and cancer. Apple Watch offers a less resource-intensive and more feasible alternative to the gold standard assessment for VO$_2$ max, indirect calorimetry, but the accuracy of its measurements remains uncertain. This study aimed to assess the validity of VO$_2$ max estimates from Apple Watch in comparison to indirect calorimetry. Thirty participants wore an Apple Watch for 5-10 days to generate a VO$_2$ max estimate. Subsequently, they underwent a maximal exercise treadmill test in accordance with the modified Åstrand protocol. The agreement between measurements from Apple Watch and indirect calorimetry was assessed using Bland-Altman analysis, mean absolute percentage error (MAPE), and mean absolute error (MAE). Overall, Apple Watch underestimated VO$_2$ max, with a mean difference of 6.07 mL/kg/min (95% CI 3.77–8.38). Limits of agreement indicated variability between measurement methods (lower -6.11 mL/kg/min; upper 18.26 mL/kg/min). MAPE was calculated as 13.31% (95% CI 10.01–16.61), and MAE was 6.92 mL/kg/min (95% CI 4.89–8.94). These findings indicate that Apple Watch VO$_2$ max estimates require further refinement prior to clinical implementation. However, further consideration of Apple Watch as an alternative to conventional VO$_2$ max prediction from submaximal exercise is warranted, given its practical utility.

## Introduction

VO$_2$ max, introduced by British physiologists Hill and Lupton, is the maximum amount of oxygen an individual can utilise during exercise [1]. It provides an evaluation of cardiorespiratory fitness and has been established as an important barometer of health [2–4]. Epidemiological evidence supports an inverse and independent association between cardiorespiratory fitness and mortality, as measured by VO$_2$ max or

**Data availability statement:** All data files are available from the primary author's GitHub repository at github.com/rorylambe/applewatch-validation

**Funding:** This project was funded by Science Foundation Ireland's National Challenge Fund (grant ID: 22/NCF/FD/10949). The funding was received by CD. The funder had no role in any aspect of the trial. Funder URL: www.sfi.ie

**Competing interests:** The authors have declared that no competing interests exist.

metabolic equivalent (MET) [5–7]. It is a predictor of cardiovascular events [8–10], and some studies report it to be more predictive of cardiovascular and all-cause mortality than well-known risk factors such as hypertension, obesity, and hypercholesterolemia [11–13]. In recognition of its significance, the American Heart Association advocates for the routine assessment of cardiorespiratory fitness and has proposed it as a clinical vital sign [14].

$VO_2$ max is the most widely accepted measure of cardiorespiratory fitness, and the gold standard for its assessment is indirect calorimetry [15]. Indirect calorimetry measures maximal oxygen uptake by recording oxygen consumption and carbon dioxide production during cardiopulmonary exercise testing (CPET) [16]. While it is the most accurate form of measurement, it is costly and resource-intensive, limiting its use in clinical practice and among the general population [17]. To overcome these difficulties, $VO_2$ max is often derived from submaximal exercise tests that require less time and are considered lower-risk [18]. Error for $VO_2$ max predictions from submaximal tests is reported to range between 1.6 and 4.1 mL/kg/min [19,20].

Wearable devices also use submaximal exercise to derive $VO_2$ max [17], and their recent proliferation has democratised monitoring of physiological measurements. They provide unobtrusive, longitudinal monitoring outside the confines of exercise physiology laboratories or healthcare settings, and may facilitate remote monitoring of cardiorespiratory fitness [21,22]. The widespread uptake of wearable devices has been catalysed by the ready availability of a suite of health-related metrics, and their increasing prominence is illustrated by the American College of Sports Medicine (ACSM), who identified wearable technology as the number one fitness trend for 2025 [23]. Over half the population in many countries now owns a wearable device – ranging from smartwatches to smart rings – with global user numbers expected to grow significantly in the coming years [24]. The wearable technology market is projected to reach a valuation of $186 billion by 2030 [25], and Apple Watch holds the largest market share with over 100 million users worldwide [26].

The first-generation Apple Watch was released in 2015, and its $VO_2$ max estimation feature was introduced with watchOS 4 in September 2017 [27]. This enabled Apple Watch Series 3 and later to estimate $VO_2$ max using data such as heart rate, Global Navigation Satellite System-derived metrics (e.g., speed, distance, and elevation), and user demographics such as sex, age, height, and weight [28]. Apple subsequently updated its $VO_2$ max algorithm in 2021 as part of watchOS 7 [28]. While this feature evoked the prospect of an assessment method that could be conducted independently of healthcare or exercise professionals, it was accompanied by notable challenges.

The accuracy of wearable-derived $VO_2$ max estimates represents one of the most significant challenges. A multitude of factors can influence measurements: physiological factors such as heart rate response to exercise and perfusion; environmental factors like skin contact and motion [29]; or external influences, including caffeine intake and carrying additional weight beyond body weight, such as bags or equipment [28]. Compounding these issues is the proprietary and evolving nature of the algorithms used by wearable devices. With each new hardware or software update, algorithms

are subject to change, necessitating recurrent validation by researchers [18,30,31]. This validation is hindered by the protracted process of conducting and publishing research. Consequently, fewer than 5% of consumer wearables have been validated for the full range of biometric outcomes they measure, including $VO_2$ max [18]. In fact, just one study evaluating the accuracy of Apple Watch for $VO_2$ max estimation has been published to date [32].

Evaluating the accuracy of $VO_2$ max estimates is fundamental to integrating wearable devices into individual health monitoring and research contexts [30]. For consumers, the reliability and validity of these measurements can inform decisions about personal health and fitness. For researchers, wearables hold the potential to revolutionise public health surveillance by enabling the collection of large-scale, population-level data without the logistical and financial constraints of laboratory-based methods [33]. On the contrary, inaccurate or inconsistent data may compromise the validity of such research, particularly when used to inform health policy or population health interventions. Validation is required to seal the fissure between technological innovation and evidence-based application in health and fitness settings.

The aim of this study was to evaluate the accuracy of $VO_2$ max estimates generated by Apple Watch in comparison to the gold standard method, indirect calorimetry. Specifically, the objectives were to assess the level of agreement between the two methods, and to quantify the degree of error in $VO_2$ max estimates from Apple Watch.

## Materials and Methods

### Study design and oversight

This cross-sectional validation study was conducted in Dublin, Ireland between December 2023 and July 2024. Recruitment commenced on Monday, 4 December 2023, and finished on Monday, 22 July 2024. Healthy adults, aged 18 or older, were recruited through social media, posters, and word of mouth. Individuals with cardiovascular or mental illness, and those using medication affecting cardiovascular function were not eligible for inclusion. All participants completed the Physical Activity Readiness Questionnaire Plus (PAR-Q+) to ensure suitability for exercise, and written informed consent was obtained. Ethical approval was granted by the University College Dublin Human Research Ethics Committee (reference number: LS-23-55) on November 15th, 2023.

### Generating $VO_2$ max estimates with Apple Watch

Apple Watch generates $VO_2$ max estimates (proprietarily referred to as 'Cardio Fitness') by measuring an individual's heart rate response to exercise during outdoor walking, running, or hiking activities on ground of less than 5% incline or decline [28]. Adequate GPS and heart rate signal must be obtained, alongside an increase of approximately 30% in heart rate from the resting value [28]. Multiple activities are required to generate an estimate, although this number varies between users [28]. Participants were informed of the required procedure and were requested to generate an estimate independently within one week of criterion testing. The process of generating a $VO_2$ max estimate was conducted in accordance with manufacturer guidelines [28]. Participants were instructed to accurately input all required demographic information in the Health app prior to completing any exercise activities, including height, weight, sex, and age. If participants did not already own an Apple Watch, they were provided with an Apple Watch Series 9 or Ultra 2 for a period of 5-10 days. All Apple Watch devices were updated to watchOS 10 or later, ensuring that this validation study exclusively assessed devices using Apple's latest $VO_2$ max prediction algorithm and the most recent software version.

### Indirect calorimetry testing

Subsequently, each participant underwent a maximal cardiopulmonary exercise test (CPET) using indirect calorimetry at the Institute for Sport and Health, University College Dublin. Indirect calorimetry is considered the gold standard for $VO_2$max testing [1,17]. An exercise treadmill test was conducted in accordance with the modified Åstrand protocol [34]. A treadmill speed between 8 and 13 km/h was selected and this remained consistent throughout [34]. The incline of the

treadmill was increased by 2.5% every two minutes following the initial three-minute warm-up period at 0% incline [34]. The COSMED Quark CPET metabolic cart (COSMED, Trentino, Italy) was calibrated prior to each test according to manufacturer instructions and testing was conducted on the h/p/cosmos Venus treadmill (h/p/cosmos, Nußdorf-Traunstein, Germany). To ensure true VO$_2$ max had been attained, participants were required to meet at least two of the following criteria: heart rate within ±10 bpm of age-predicted maximum (220 – age); respiratory exchange ratio (RER) ≥ 1.15; rate of perceived exertion (RPE) ≥ 17; VO$_2$ plateau, defined as an increase in VO$_2$ of less than 150 mL/min/kg, with an increase in work rate as evidenced by physiological data [35]. If at least two of these criteria were not met, the value was regarded as a VO$_2$ peak. Participants were instructed to refrain from caffeine and nicotine for 12 hours prior to testing, and to avoid strenuous activity and alcohol for a minimum of 24 hours [15].

### Outcomes

The primary outcome was the agreement between VO$_2$ max estimates derived from Apple Watch with those obtained via indirect calorimetry, which was calculated using Bland-Altman limits of agreement analysis [36], mean absolute percentage error (MAPE), and mean absolute error (MAE).

### Statistical analysis

The target sample size was calculated based on the sample sizes of previous studies investigating the measurement accuracy of other consumer wearable devices [32,37,38]. Additional participants were not recruited to account for potential dropouts due to the cross-sectional nature of the study design. Data from the COSMED Quark CPET were filtered using a time average of 30 seconds and exported to Microsoft Excel. The highest time-averaged VO$_2$/kg value was interpreted as VO$_2$ max. This was compared to the most recent Apple Watch VO$_2$ max estimate available in the Health app. Participants who did not attain VO$_2$ max were not included in the final analysis.

To evaluate the agreement between Apple Watch and the criterion, Bland-Altman limits of agreement analysis was conducted. The mean difference (bias) and 95% limits of agreement (LoA), defined as the mean difference ±1.96 times the standard deviation of the differences, were calculated. A Bland-Altman plot was generated to visually evaluate the agreement. Mean absolute percentage error (MAPE) and mean absolute error (MAE) were also calculated. The t-critical values were used to compute corresponding 95% confidence to account for the sample size. Analyses were conducted in Python (version 3.12) using pandas, Matplotlib, and NumPy packages. The scripts used are available at github.com/rorylambe/applewatch-validation.

## Results

### Baseline characteristics

A total of 30 individuals participated in this study (mean age [SD], 31.86 [13.99] years; 50% female). Of the 30 participants, two (one male, one female) did not achieve the required criteria for VO$_2$ max, and as such 28 participants were included in the analyses. One of the excluded participants met only the threshold for rate of perceived exertion whereas the other failed to attain any of the criteria. Both excluded participants were wearing Apple Watch Series 9. The cardiorespiratory fitness level of each participant, determined by indirect calorimetry testing, was classified in accordance with the reference standard from the Fitness Registry and the Importance of Exercise National Database (FRIEND) [39]. 21 participants were classified as having either 'Superior' or 'Excellent' cardiorespiratory fitness, while nine individuals were classified as 'Good' or 'Fair'. Participant baseline characteristics, including BMI and Fitzpatrick skin tone, are listed in Table 1.

### Apple Watch agreement with the criterion

Overall, Apple Watch underestimated VO$_2$ max in comparison to indirect calorimetry. The mean difference was 6.07 mL/kg/min (SD 6.22; 95% confidence interval [CI] 3.77–8.38). Bland-Altman limits of agreement (LoA) indicated variability

**Table 1. Characteristics of the participants at baseline.**

| Characteristic | Value |
|---|---|
| Age, mean (SD), years | 31.86 (13.99) |
| Female, no. (%) | 15 (50%) |
| Height, mean (SD), cm | 172.23 (7.85) |
| Weight, mean (SD), kg | 70.55 (9.13) |
| BMI, mean (SD), kg/m$^2$ | 23.76 (2.54) |
| Fitzpatrick skin tone [a] | |
| Type I | 4 |
| Type II | 22 |
| Type III | 4 |
| CRF level [b] | |
| Superior, no. | 3 |
| Excellent, no. | 18 |
| Good, no. | 8 |
| Fair, no. | 1 |
| Apple Watch model [c] | |
| Series 9 | 17 |
| Ultra 2 | 6 |
| Series 8 and SE 2 | 2 of each |
| Series 5, 6, 7 | 1 of each |

BMI: Body Mass Index

[a] Number of participants of each skin tone according to the Fitzpatrick Scale.

[b] Number of participants of each cardiorespiratory fitness level classified according to FRIEND.

[c] Number of participants who wore each Apple Watch model during the study.

between the two measurement methods (lower LoA -6.11 mL/kg/min; upper LoA 18.26 mL/kg/min). The Bland-Altman plot is presented in Fig 1. The mean absolute percentage error (MAPE) was 13.31% (95% CI 10.01–16.61); mean absolute error (MAE) was 6.92 mL/kg/min (95% CI 4.89–8.94). To assess statistical power and effect size, a one-sample t-test was conducted comparing Apple Watch and criterion VO$_2$ max values. This analysis yielded a t-value of –5.07 (df = 27, p < 0.001), indicating a significant difference with a large effect size (Cohen's d = –0.96). A post-hoc power analysis (α = 0.05) confirmed the sample size (n = 28) provided >99% power, suggesting the study was sufficiently powered to detect a true difference between the measurement methods, thereby minimising the risk of Type II error. No trends in accuracy could be observed based on Apple Watch model. Results are summarised in Table 2.

## Discussion

This validation study evaluated the accuracy of VO$_2$ max estimates from Apple Watch compared to the gold-standard method of indirect calorimetry. Overall, Apple Watch underestimated VO$_2$ max, with a mean difference of 6.07 mL/kg/min (SD 6.22) and a MAPE of 13.31%. Bland-Altman limits of agreement indicated variability between the two measurement methods (lower -6.11 mL/kg/min; upper 18.26 mL/kg/min).

These findings align with those of a prior study evaluating Apple Watch Series 7, which reported a similar mean underestimation of 4.5 mL/kg/min [32]. However, important methodological differences exist. In the previous study, criterion VO$_2$ max values were obtained through a graded exercise test conducted on a cycle ergometer, rather than using a treadmill-based test. Apple Watch derives VO$_2$ max from walking, running or hiking activities [28], and comparison of VO$_2$

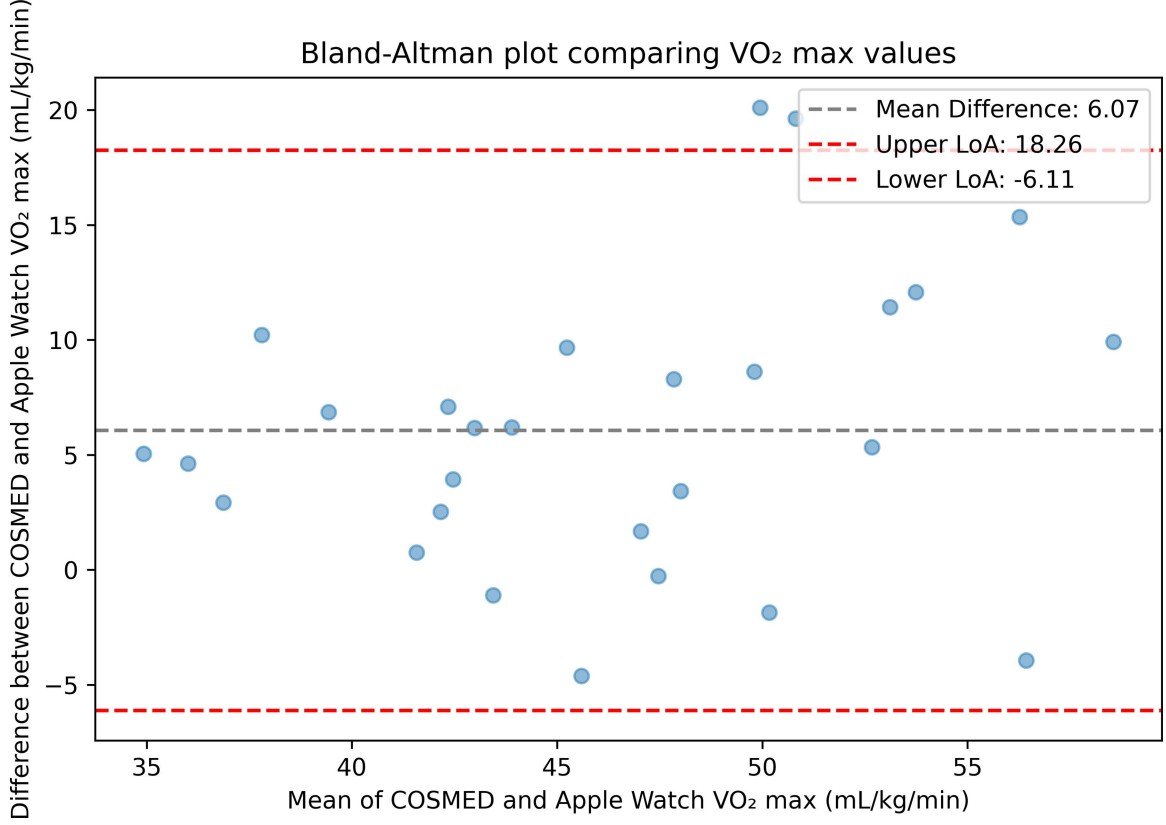

**Fig 1. Bland-Altman plot demonstrating the agreement between VO2 max measurements from Apple Watch and the criterion measure.** Upper LoA = upper limit of agreement; Lower LoA = lower limit of agreement.

**Table 2. Statistical measures of agreement between Apple Watch and indirect calorimetry.**

| Statistical measure | Result |
|---|---|
| Mean (SD), mL/kg/min | |
| Apple Watch | 43.27 (6.34) |
| COSMED* | 49.35 (7.79) |
| Standard error of the mean, mL/kg/min | |
| Apple Watch | 1.20 |
| COSMED | 1.47 |
| Standard deviation of the differences, mL/kg/min | 6.22 |
| Mean difference (95% CI), mL/kg/min | 6.07 (3.77–8.38) |
| Bland-Altman limits of agreement (lower, upper), mL/kg/min | −6.11, 18.26 |
| Mean absolute percentage error (95% CI) | 13.31% (10.01–16.61) |
| Mean absolute error, mL/kg/min (95% CI) | 6.92 (4.89–8.94) |

\* COSMED = COSMED Quark CPET metabolic cart.

max measured through different exercise modalities – such as cycling compared with running – is not advised due to the inherent discrepancies in results they produce [40]. Typically, treadmill tests yield higher $VO_2$ max values, although the difference is smaller in well-trained cyclists [41]. This difference underpins the INTERLIVE consortium's recommendation that criterion testing methods should closely align with the activity used by the wearable device to generate $VO_2$ max predictions [17].

While these methodological differences are important to consider, the authors noted distinct trends based on the fitness levels of participants [32]. For individuals with good or excellent fitness, Apple Watch demonstrated a propensity to underestimate $VO_2$ max, whereas among those with poor fitness, there was a tendency to overestimate. Although these authors found an overall underestimation, a separate meta-analysis of fourteen studies, which included multiple wearable device brands, reported a minor overestimation (pooled bias -0.09 mL/kg/min) [17]. Despite a very low bias, the limits of agreement were comparatively wide (lower -9.92 mL/kg/min; upper 9.74 mL/kg/min), suggesting both positive and negative differences in estimates of a relatively large magnitude, which collectively yielded a bias close to zero. A propensity to both under- and overestimate may arise from prediction algorithms that are developed using population-level data or mean values for specific demographics. Given our study sample predominantly consisted of individuals with high cardiorespiratory fitness levels, this may constitute one explanation for our finding of underestimation. Importantly, it highlights the need to consider participant demographic characteristics, as well as multiple statistical measures of agreement, when interpreting results.

Our findings, combined with the broader literature, underscore the challenge of deriving maximal oxygen consumption from submaximal exercise for wearable devices. Like conventional prediction equations, Apple's prediction algorithm is founded on heart rate response to exercise [28]. The inclusion of machine learning techniques and additional motion sensor measurements is also probable, although the precise nature of the algorithm remains undisclosed [30]. In clinical settings, submaximal exercise testing is often used alongside prediction equations to estimate $VO_2$ max, owing to safety and feasibility concerns [35]. These equations predict $VO_2$ max based on an individual's physiological response to exercise of a quantified workload, and six assumptions – outlined by the ACSM – must be achieved to ensure optimal accuracy [35]. They include a minimal difference between actual and predicted maximal heart rate, knowledge of medication or substances affecting heart rate, and the existence of a linear relationship between heart rate and work rate. Yet, accuracy from traditional prediction equations can vary significantly [42–44]. A study (n = 541) evaluating three different submaximal exercise tests for estimating maximal oxygen uptake reported a standard error of the estimation ranging from 3.7 to 4.5 mL/kg/min [45]. In a separate study examining the Chester Step Test, the predicted aerobic capacity had a standard error of 3.9 mL/kg/min [19,46]. Comparatively, measurement error for laboratory-based indirect calorimetry has been estimated at ±5%, with a meta-analysis of 39 studies reporting a mean standard error of 2.58 mL/kg/min [40,47]. Considering this literature, the results of our validation study indicate that Apple Watch $VO_2$ max predictions fall approximately two to three standard deviations beyond the typical error of the gold standard measurement. However, they align more closely with the magnitude of error reported for traditional $VO_2$ max predictions derived from submaximal exercise.

While machine learning and continuous monitoring are distinct merits of Apple Watch compared to traditional prediction methods, the uncontrolled nature of the exercise used to generate Apple Watch $VO_2$ max estimates presents a considerable challenge. Limited information of external factors that affect heart rate response to exercise – including exercise surface, environmental temperature, or alcohol intake – can adversely impact predictions [28]. Moreover, heart rate response to lower-intensity exercise – often used to estimate $VO_2$ max – varies considerably between individuals [48]. Estimates based on such exercise may misinform predictions. Due to the proprietary nature of Apple's algorithm, we remain unaware of the specific data, or features, used to train and develop the prediction model. Therefore, we cannot be certain what factors most influence accuracy, or in which instances predictions are most reliable [30]. Considering this, a defined testing protocol – requiring users to input specific information and conduct a particular type of exercise – may reduce the level of uncertainty and error. Despite the imprecision of predictions from submaximal exercise – both from traditional testing methods and wearable devices – its widespread clinical use illustrates its value. The potential for Apple Watch to provide

a commensurate assessment method that can be conducted independently by members of the general public warrants careful consideration. To investigate this, validation of contemporary technology is apt.

Continual, or 'living', validation is essential to keep pace with the iterative commercial ecosystem. The annual update cycle of Apple Watch poses a significant challenge to evaluating current technology, and the prolonged academic publication process often means that the hardware or software under validation has been discontinued by the time of publication [18,30]. This is accentuated by the rapid advancement of machine learning – and the associated updates to watchOS – which play an increasingly important role in providing accurate measurements due to the complexity of interpreting multiple sensor measurements and sensitive photoplethysmography waveforms [49,50]. Amidst these challenges, this study provides a timely evaluation of Apple's most recent optical heart rate sensor and its latest $VO_2$ max prediction algorithm [28,51]. It is the first study to evaluate Apple Watch Series 9 and Ultra 2. As wearable technology evolves, agile validation will be critical to realising the potential of consumer wearables in large-scale public health applications and personalised fitness monitoring.

Central to this potential is the self-directed nature of wearable device measurements, and this study's methodology was designed to reflect this. Participants were not required to adhere to a prescribed protocol to generate a $VO_2$ max prediction. Rather, they were solely informed of the requirements, meaning that estimates were generated through activities that more accurately reflected each individual's real-world use, enhancing the ecological validity of our findings. Additionally, a robust statistical approach was used, guided by recommendations of the INTERLIVE expert consortium [17], and a sex-balanced study sample was recruited. However, the predominance of participants with high levels of cardiorespiratory fitness represents the study's most significant limitation. Only one participant was classified as having 'Fair' cardiorespiratory fitness, while all others were categorised as 'Good', 'Excellent', or 'Superior'. This limits the generalisability of our findings to populations with lower fitness levels, including clinical cohorts. Additionally, the sample primarily consisted of younger individuals, despite efforts to recruit participants across a broad age range. Another limitation is the uncontrolled procedure of generating Apple Watch $VO_2$ max estimates. While this protocol reflects real-world use, it introduces unquantified external variables that may have influenced prediction accuracy. Lastly, only one $VO_2$ max estimate was collected per participant, preventing analysis of intra-subject variability, which may have provided insights into reliability and changes in measurement accuracy over time.

The findings of this study have important implications for the use of wearable devices such as Apple Watch in clinical and research contexts. $VO_2$ max estimates from Apple Watch offer a cost-effective and scalable alternative to laboratory-based testing, enabling studies involving diverse populations, as well as digital endpoints for clinical trials. However, given the observed variability and underestimation of $VO_2$ max in this study, caution is warranted when interpreting the data in clinical or research settings. While Apple Watch may provide a suitable estimation of aerobic capacity for general fitness monitoring, our findings suggest that estimates are not sufficiently accurate to inform clinical decision-making. Future research should include individuals with diverse cardiorespiratory fitness levels to enhance generalisability, support the refinement of predictive algorithms, and develop frameworks for ongoing evaluation of wearable devices. Such efforts are essential to bridging the gap between technological innovation and evidence-based practice, ultimately enhancing the utility of wearable devices for individual health monitoring and public health applications.

## Conclusions

This study found that Apple Watch underestimated $VO_2$ max compared to indirect calorimetry in a sample predominantly composed of individuals with high cardiorespiratory fitness levels. The margin of error more closely aligned with that of $VO_2$ max derived from conventional submaximal exercise testing than with $VO_2$ max obtained via indirect calorimetry. The unstructured nature of the exercise used to generate Apple Watch $VO_2$ max predictions introduces external factors – such as environmental conditions and variations in heart rate response to exercise – that are not fully accounted for by the device. Nevertheless, Apple Watch holds promise as a practical and accessible alternative to conventional submaximal

exercise testing. Its potential clinical utility warrants further investigation, and ongoing validation is required to evaluate its accuracy as hardware and software evolve. Future research should focus on expanding validation efforts to include diverse populations, ensuring these devices can be effectively integrated into both individual health monitoring and broader public health applications.

## Author contributions

**Conceptualization:** Rory Lambe, Ben O'Grady, Maximus Baldwin, Cailbhe Doherty.

**Data curation:** Rory Lambe, Ben O'Grady, Maximus Baldwin, Cailbhe Doherty.

**Formal analysis:** Rory Lambe.

**Funding acquisition:** Cailbhe Doherty.

**Investigation:** Rory Lambe, Ben O'Grady, Cailbhe Doherty.

**Methodology:** Rory Lambe, Ben O'Grady, Maximus Baldwin, Cailbhe Doherty.

**Project administration:** Rory Lambe, Ben O'Grady, Maximus Baldwin, Cailbhe Doherty.

**Software:** Rory Lambe.

**Supervision:** Cailbhe Doherty.

**Validation:** Rory Lambe, Ben O'Grady, Maximus Baldwin.

**Visualization:** Rory Lambe.

**Writing – original draft:** Rory Lambe, Ben O'Grady, Maximus Baldwin, Cailbhe Doherty.

**Writing – review & editing:** Rory Lambe, Maximus Baldwin, Cailbhe Doherty.

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
