## [Decision Letter · Decision Letter 0]

25 Feb 2025

Dear Dr. Lambe,

Thank you for submitting your manuscript to PLOS ONE. After careful consideration, we feel that it has merit but does not fully meet PLOS ONE’s publication criteria as it currently stands. Therefore, we invite you to submit a revised version of the manuscript that addresses the points raised during the review process.

**ACADEMIC EDITOR:**plosone@plos.org . A rebuttal letter that responds to each point raised by the academic editor and reviewer(s). You should upload this letter as a separate file labeled 'Response to Reviewers'.A marked-up copy of your manuscript that highlights changes made to the original version. You should upload this as a separate file labeled 'Revised Manuscript with Track Changes'.An unmarked version of your revised paper without tracked changes. You should upload this as a separate file labeled 'Manuscript'.

We look forward to receiving your revised manuscript.

Kind regards,

Emiliano Cè, Ph.D.

Academic Editor

PLOS ONE

Journal Requirements:

3. Please ensure that you refer to Figure 2 in your text as, if accepted, production will need this reference to link the reader to the figure.

Reviewers' comments:

Reviewer's Responses to Questions

**Comments to the Author**

1. Is the manuscript technically sound, and do the data support the conclusions?

Reviewer #1: Partly

Reviewer #2: Yes

2. Has the statistical analysis been performed appropriately and rigorously?

Reviewer #1: Yes

Reviewer #2: Yes

3. Have the authors made all data underlying the findings in their manuscript fully available?

Reviewer #1: Yes

Reviewer #2: Yes

4. Is the manuscript presented in an intelligible fashion and written in standard English?

Reviewer #1: Yes

Reviewer #2: Yes

Reviewer #1: The study compares the accuracy of Apple Watch VO₂ max predictions to indirect calorimetry, in order to address a crucial problem in wearable health technologies. The topic is extremely pertinent in light of the growing use of consumer wearable technology for health monitoring. The backdrop is organized effectively, highlighting the significance of VO₂ max as a health indicator and recognizing the shortcomings of the available evaluation techniques.

The testing procedure, exclusion criteria, and participant recruiting are all covered in depth in the methodology's extensive documentation. But there are a few issues:

- The sample size (n=30, of which 28 were analyzed) is too small to extrapolate results.

- The study's relevance to populations with lower fitness levels may be limited because it mostly involves people with strong cardiorespiratory fitness.

- There is no explicit explanation in the study as to why particular Apple Watch models were chosen over others or whether hardware variations affected the findings.

The statistical approach is relative good, employing Bland-Altman analysis, mean absolute percentage error (MAPE), and mean absolute error (MAE) to assess agreement. However, the study could have benefited from additional analysis:

- A regression model to explore factors influencing the discrepancies.

- Subgroup analysis to determine whether the accuracy of estimates varies by demographic factors (e.g., age, sex, fitness level).

The authors appropriately cite relevant literature, including a meta-analysis on wearable-derived VO₂ max estimates. They also highlight the proprietary nature of Apple’s prediction algorithm as a limitation, a crucial consideration in wearable validation research. The study suggests that Apple Watch could serve as an alternative to traditional submaximal exercise testing but requires further refinement. However, the discussion could be improved by:

- Addressing whether firmware updates might influence VO₂ max accuracy over time.

- Providing a clearer stance on whether Apple Watch is suitable for specific use cases (e.g., general fitness tracking vs. clinical decision-making).

While the study contributes valuable insights, its limitations—small sample size, fitness level bias, and high variability in agreement—raise concerns about its broader applicability.

With above suggestions, the paper would be more robust and suitable for publication.

Reviewer #2: I would like to applaud the authors on this work, it is important to scrutinize the quality of data coming out of "consumer devices" and we need more research focus on this topic. I only have a few suggestions for the authors, some I would consider "major" and would need to be addressed for publication and some I would consider optional. Overall, I think the manuscript can be improved with minimal effort and will be suitable for publication.

Major suggestions:

(1) Fix your figures. I see no purpose for Fig 1 as it stands, the data from it is actually in Fig 2 and the graphics are neither discussed nor referenced. Fig 2 is also not referenced anywhere in the manuscript. And please add some captions. The figures are probably the weakest part of this manuscript.

(2) Bring some data / discussion into the manuscript around the repeatability and/or within-subject variance of the gold standard test. It would have been great if the authors collected more than 1 VO2 MAX estimate / subject to be able to derive this from their own data, however, I would be happy with just reference values from published literature. And then the authors should compare the Apple Watch bias / precision / accuracy to the variance of the gold standard test. For example, I would really want to know: is 6 mL/kg/min difference within 1,2,3 SDs of the gold standard test?

(3) Include all key parameters that Apple uses in their algo. For example, "sex" and "height" are missing on page 4, line 84 and "sex" is missing on page 6, line 140.

(4) Page 7, in the "Outcomes" section of the Methods, provide a reference to the Bland and Altman work, for example: Altman DG, Bland JM. Measurement in medicine: the analysis of method comparison studies. Statistician. 1983;32:307–17. 10.2307/2987937 or Bland JM, Altman DG. Statistical methods for assessing agreement between two methods of clinical measurement. Int J Nurs Stud. 2010;47:931–6. 10.1016/j.ijnurstu.2009.10.001

Minor suggestions:

(1) Include some data from other submaximal exercise tests (mentioned on page 3, second paragraph) and discuss how other tests compare to the Apple Watch estimate.

(2) Calculate not only the bias difference across testing methods but also the precision and create BA plots for those too.

**Do you want your identity to be public for this peer review?** For information about this choice, including consent withdrawal, please see our Privacy Policy

Reviewer #1: No

Reviewer #2: **Yes: ** Jordan Brayanov

---

## [Author Response · Author response to Decision Letter 1]

4 Mar 2025

Response to Academic Editor

COMMENT: 1. Please ensure that your manuscript meets PLOS ONE's style requirements, including those for file naming. The PLOS ONE style templates can be found at https://journals.plos.org/plosone/s/file?id=wjVg/PLOSOne_formatting_sample_main_body.pdf and https://journals.plos.org/plosone/s/file?id=ba62/PLOSOne_formatting_sample_title_authors_affiliations.pdf

RESPONSE: Font size for headings and subheadings has been altered, in accordance with style and formatting requirements. A caption has been inserted for Figure 1 — please note that Fig 1 in this submission is the figure that was named Figure 2 in the previous draft. Figure 1 from the previous submission has been removed in response to the comments of Reviewer 2.

Sentence case has been implemented for the title, and the Acknowledgements section has been reformatted. File names have also been altered.

--

COMMENT: 2. Your ethics statement should only appear in the Methods section of your manuscript. If your ethics statement is written in any section besides the Methods, please delete it from any other section.

RESPONSE: The ethics statement has been removed from the Acknowledgements section, and now only appears in the Methods section.

--

COMMENT: 3. Please ensure that you refer to Figure 2 in your text as, if accepted, production will need this reference to link the reader to the figure.

RESPONSE: In this submission, there is now just one figure — in response to the comments of Reviewer 2. A reference to Figure 1 has been included in the text (line 254, under sub-heading ‘Apple Watch agreement with the criterion’):

“The Bland-Altman plot is presented in Fig 1.”

--

Response to Reviewer 1

COMMENT: The study compares the accuracy of Apple Watch VO₂ max predictions to indirect calorimetry, in order to address a crucial problem in wearable health technologies. The topic is extremely pertinent in light of the growing use of consumer wearable technology for health monitoring. The backdrop is organized effectively, highlighting the significance of VO₂ max as a health indicator and recognizing the shortcomings of the available evaluation techniques.

RESPONSE: Thank you for your constructive comments highlighting the importance of this topic and its growing pertinence. I’m grateful for your feedback as it has helped us to improve this paper.

COMMENT: The testing procedure, exclusion criteria, and participant recruiting are all covered in depth in the methodology's extensive documentation. But there are a few issues:

- The sample size (n=30, of which 28 were analyzed) is too small to extrapolate results.

- The study's relevance to populations with lower fitness levels may be limited because it mostly involves people with strong cardiorespiratory fitness.

RESPONSE: Thank you for recognising the detail of this manuscript’s methodology. We certainly agree that the size of the sample — and the predominance of individuals with high level of cardiorespiratory fitness — limits the extrapolation of our results to the general population. A larger sample size including individuals with a wide range of cardiorespiratory fitness would greatly enhance the generalisability of our findings, and we acknowledge this as an important limitation in the Discussion section of the manuscript.

We have now conducted a post-hoc power analysis for our sample size, using the following information from our results:

• Apple Watch mean VO2 max (standard deviation): 43.27 (6.34)

• Criterion mean VO2 max: 49.35

• Sample size: 28

We calculated the following results from the post-hoc power analysis:

• Standard Error: 1.198

• t value: -5.075

• p-value: 0.00002

• Cohen's d: -0.959

• Power: 0.998

The results of this power analysis suggest our study is well-powered to detect an effect. We have now added this analysis to the Results section of our manuscript:

“To assess statistical power and effect size, a one‐sample t‐test was conducted comparing Apple Watch and criterion VO₂ max values. This analysis yielded a t‐value of –5.07 (df = 27, p < 0.001), indicating a significant difference with a large effect size (Cohen’s d = –0.96). A post-hoc power analysis (α = 0.05) confirmed the sample size (n = 28) provided >99% power, suggesting the study was sufficiently powered to detect a true difference between the measurement methods, thereby minimising the risk of Type II error.”

While a concerted effort was made by our research team to recruit individuals across a wide range of age and fitness levels, this recruitment proved challenging. Due to the exhaustive nature of a maximal exercise test, few individuals with low levels of cardiorespiratory fitness levels were willing to participate, and all included participants aged 55-65 had ‘Excellent’ or ‘Superior’ cardiorespiratory fitness. Previously, just one study evaluating the accuracy of VO2 max measurements from Apple Watch has been published. That study compared predictions from Apple Watch Series 7 to VO2 max values obtained during a cycling protocol, measured using a portable gas analyser (n=19, 7 female). This study builds on the current literature base by providing the first assessment of Apple Watch Series 9 and Ultra 2, in a larger sex-balanced cohort. This is the first study to compare Apple Watch VO2 max measurements to indirect calorimetry — regarded as the criterion standard — which were obtained during an exercise treadmill test, closely mirroring the activity required to generate an Apple Watch prediction. This is in accordance with guidelines for validation published by the expert-led INTERLIVE consortium.

--

COMMENT: There is no explicit explanation in the study as to why particular Apple Watch models were chosen over others or whether hardware variations affected the findings.

RESPONSE: Thank you for highlighting this important point. At the time of data collection, the most recently released models were Apple Watch Series 9 and Ultra 2. The Series 9 has recently been replaced by Series 10, while the Ultra 2 remains the flagship model. A substantial number of these devices were purchased to provide to participants (17 x €449 for Series 9, 6 x €899 for Ultra 2) so that the latest device could be validated. Due to funding restraints, and in an effort to recruit a greater number of participants, the authorship team also recruited individuals who owned an Apple Watch which was capable of running watchOS 10 or later. In this way, all models included in this validation study were running the latest version of watchOS software, and all devices used the latest algorithm to generate a VO2 max prediction.

The following information has been added in the Methods section, under the sub-heading ‘Generating VO2 max estimates with Apple Watch’:

“If participants did not already own an Apple Watch, they were provided with an Apple Watch Series 9 or Ultra 2 for a period of 5-10 days to generate an estimate. All Apple Watch devices were updated to watchOS 10 or later, ensuring that this validation study exclusively assessed devices using Apple’s latest VO2 max prediction algorithm and the most recent software version.”

No trends in accuracy could be observed based on the individual Apple Watch model tested, nor based on participant demographic (please also see our response to the following comments). This may indicate that the prediction algorithm has a large role in determining the accuracy of VO2 max estimates, compared to differences in hardware. To facilitate transparent reporting of results, we have uploaded additional data relating to Apple Watch model and participant demographic to github.com/rorylambe/applewatch-validation.

We have also added additional information about hardware variations to the Results section of the manuscript.

--

COMMENT: The statistical approach is relative good, employing Bland-Altman analysis, mean absolute percentage error (MAPE), and mean absolute error (MAE) to assess agreement. However, the study could have benefited from additional analysis:

- A regression model to explore factors influencing the discrepancies.

- Subgroup analysis to determine whether the accuracy of estimates varies by demographic factors (e.g., age, sex, fitness level).

RESPONSE: This was a very thought-provoking comment, and the authorship team spent much time examining possibilities for further analyses. We designed the statistical analysis plan of this study in accordance with the expert-led INTERLIVE consortium’s recommendations. To explore the potential influence of these factors, we have conducted two linear regression models — one each for age, and sex. Regression was chosen as a statistically robust approach that adjusts for cofounders and retains sample size.

To reduce a risk of overfitting and to maximise statistical power, we chose to run separate simple linear regression models for sex and age rather than a multiple regression model. Despite this, both models remain statistically underpowered and prone to collinearity. As a consequence, having conducted these regression analyses, we feel that it may be more suitable not to include these underpowered analyses in the final manuscript. Additional analyses were not part of the initial statistical analysis plan, and the authorship team do not believe that sex or age is likely to have a substantial influence given the nature of the cohort recruited by Apple in their development of their proprietary algorithm.

Neither regression models revealed a statistically significant association. For sex, we found that the level of agreement between Apple Watch and the criterion measurement was greater for males, although this was not statistically significant. Sex accounted for 12.7% of the variance in discrepancy between measurement methods. Age did not demonstrate a statistically significant association, and it accounted for just 2% of variance.

Although a regression model for fitness level would be of substantial value, we determined that it would be inappropriate to conduct in this instance. This is due to the nature of the spread of cardiorespiratory fitness levels across each category, and the resulting number of individuals per category. There is an inadequate number of participants in each category for sufficient statistical power, and to account for collinearity.

We have, however, prepared an updated version of the Results sections of the manuscript to include if the Academic Editor deems these regression models to be of additional value to the paper.

Results:

Overall, Apple Watch underestimated VO2 max in comparison to indirect calorimetry. The mean difference was 6.07 mL/kg/min (SD 6.22; 95% confidence interval [CI] 3.77–8.38). Bland-Altman limits of agreement (LoA) indicated variability between the two measurement methods (lower LoA -6.11 mL/kg/min; upper LoA 18.26 mL/kg/min). The Bland-Altman plot is presented in Fig 1. The mean average percentage error (MAPE) was 13.31% (95% CI 10.01–16.61); mean absolute error (MAE) was 6.92 mL/kg/min (95% CI 4.89–8.94). To assess statistical power and effect size, a one‐sample t‐test was conducted comparing Apple Watch and criterion VO₂ max values. This analysis yielded a t‐value of –5.07 (df = 27, p < 0.001), indicating a significant difference with a large effect size (Cohen’s d = –0.96). A post-hoc power analysis (α = 0.05) confirmed the sample size (n = 28) provided >99% power, suggesting the study was sufficiently powered to detect a true difference between the measurement methods, thereby minimising the risk of Type II error. No trends in accuracy could be observed based on Apple Watch model. Results are summarised in Table 2. “Two separate linear regression models were used to examine the associations of sex and age (predictor variables in respective models), with the signed difference in VO2 max values between Apple Watch and the criterion (dependent variable in each model). Neither sex nor age demonstrated a statistically significant association with measurement accuracy. Sex accounted for 12.7% of the variance in discrepancy between measurement methods (β = -4.359, SE = 2.237; t = -1.949; 95% CI: -8.956–0.239; R2 = 0.127; F = 3.797). The coefficient for male sex indicated that the level of agreement was, on average, 4.36 mL/kg/min higher in males compared to females, however, this was not statistically significant (p = 0.062). Age explained just 2% of variance and was not a significant predictor (β = 0.0623, SE = 0.085; t = 0.734; 95% CI: -0.112–0.237; R2 = 0.02; F = 0.539). It was deemed inappropriate to conduct regression based on cardiorespiratory fitness due to the imbalanced spread of participants across each category. Additionally, no trends in accuracy could be observed based on Apple Watch model. Raw results data, along with demographic information used for regression, has been published at github.com/rorylambe/applewatch-validation.”

--

COMMENT: The authors appropriately cite relevant literature, including a meta-analysis on wearable-derived VO₂ max estimates. They also highlight the proprietary nature of Apple’s prediction algorithm as a limitation, a crucial consideration in wearable validation research. The study suggests that Apple Watch could serve as an alternative to traditional submaximal exercise testing but requires further refinement. However, the discussion could be improved by:

- Addressing whether firmware updates might influence VO₂ max accuracy over time.

RESPONSE: In response to your comments, we have included your suggestions in the Discussion.

Hardware and software both influence the accuracy of VO2 max estimates. Advancements in photoplethysmography sensors — in particular increases in light frequency and intensity — have yielded improvements in heart rate accuracy, during high-intensity activities especially. These heart rate measurements form a critical element of prediction algorithms for VO2 max. A growing source of improvement in the accuracy of these predictions, however, is the machine learning (ML) that underpins prediction algorithms. Rapid advancements in ML are facilitating improved recognition of motion artefacts, and detection of important segments in photoplethysmography waveforms. Consequently, firmware updates will have a substantial impact on accuracy, however, significant updates can only be achieved by the development of new machine learning algorithms which requires the collection of substantial amounts of data. This is one reason we do not see yearly updates to Apple’s VO2 max prediction algorithm.

We have added the following information to the Discussion:

“The annual update cycle of Apple Watch poses a significant challenge to evaluating current technology, and the prolonged academic publication process often means that the hardware or software under validation has been discontinued by the time of publication [18, 28]. This is accentuated by the rapid advancement of machine learning — and the associated updates to watchOS — which play an increasingly important role in providing accurate measurements due to the complexity of interpreting multiple sensor measurements and sensitive photoplethysmography waveforms [43, 44]. Amidst these challenges, this study provides a timely evaluation of Apple’s most recent optical heart rate sensor and its latest VO₂ max prediction algorithm [26, 45].”

--

COMMENT: Providing a clearer stance on whether Apple Watch is suitable for specific use cases (e.g., general fitness tracking vs. clinical decision-making).

RESPONSE: We acknowledge the importance of providing a clear interpretation of our results, and their implications for use cases. To address your comment, we have added additional information to the Discussion. In doing so, we aim to clarify that Apple Watch VO2 max predictions do not demonstrate sufficient accuracy to inform clinical decisions, based on our findings. Although estimates are accompanied by a degree of inaccuracy, Apple Watch may provide an accessible estimation of aerobic capacity for the purpose of general fitness monitoring.

The following amendments have been made:

“VO₂ max estimates from Apple Watch offer a cost-effective and scalable alternative to traditional laboratory-based testing, enabling st

---

## [Decision Letter · Decision Letter 1]

5 Apr 2025

Dear Dr. Lambe,

Thank you for submitting your manuscript to PLOS ONE. After careful consideration, we feel that it has merit but does not fully meet PLOS ONE’s publication criteria as it currently stands. Therefore, we invite you to submit a revised version of the manuscript that addresses the points raised during the review process.

**ACADEMIC EDITOR: **plosone@plos.org . A rebuttal letter that responds to each point raised by the academic editor and reviewer(s). You should upload this letter as a separate file labeled 'Response to Reviewers'.A marked-up copy of your manuscript that highlights changes made to the original version. You should upload this as a separate file labeled 'Revised Manuscript with Track Changes'.An unmarked version of your revised paper without tracked changes. You should upload this as a separate file labeled 'Manuscript'.

We look forward to receiving your revised manuscript.

Kind regards,

Emiliano Cè, Ph.D.

Academic Editor

PLOS ONE

Journal Requirements:

Reviewers' comments:

Reviewer's Responses to Questions

**Comments to the Author**

Reviewer #1: (No Response)

Reviewer #2: All comments have been addressed

2. Is the manuscript technically sound, and do the data support the conclusions?

Reviewer #1: Yes

Reviewer #2: Yes

3. Has the statistical analysis been performed appropriately and rigorously?

Reviewer #1: Yes

Reviewer #2: Yes

4. Have the authors made all data underlying the findings in their manuscript fully available?

Reviewer #1: Yes

Reviewer #2: Yes

5. Is the manuscript presented in an intelligible fashion and written in standard English?

Reviewer #1: Yes

Reviewer #2: Yes

Reviewer #1: The revised manuscript presents a well-structured and thorough validation study on the accuracy of Apple Watch VO₂ max estimates compared to the gold-standard method of indirect calorimetry. The authors have effectively addressed the concerns raised in the initial review by improving methodological transparency, incorporating additional statistical analyses, and expanding the discussion to provide better context for their findings. The inclusion of a post-hoc power analysis strengthens the study’s credibility, confirming that the sample size of 28 participants was statistically sufficient to detect meaningful differences between Apple Watch and indirect calorimetry. Additionally, the authors conducted regression analyses to assess the influence of age and sex on measurement discrepancies, although no significant associations were found. These additions enhance the depth of the study and demonstrate a rigorous approach to validation.

A major improvement in the revised version is the expanded discussion, which now situates Apple Watch’s performance within the broader context of wearable technology and traditional submaximal exercise tests. The authors acknowledge the proprietary nature of Apple’s algorithm and how software updates may influence accuracy over time. Furthermore, they compare the observed error margins to those of conventional submaximal VO₂ max prediction methods, reinforcing the notion that Apple Watch may be more suitable for general fitness tracking than for clinical decision-making. The discussion also now includes a reference to the inherent variability of indirect calorimetry (~2.58 mL/kg/min), providing a useful benchmark for interpreting the Apple Watch’s underestimation of VO₂ max by 6.07 mL/kg/min.

Despite these strengths, some limitations remain. The sample is still skewed towards individuals with high cardiorespiratory fitness, which restricts the generalizability of findings to less fit or clinical populations. Although the authors acknowledge this in the discussion, future studies should aim for a more diverse participant pool. Another limitation is the uncontrolled nature of the Apple Watch data collection process, as participants generated VO₂ max estimates independently over several days without standardized exercise conditions. While this enhances ecological validity, it introduces external variables that may have influenced accuracy. Additionally, only one Apple Watch measurement was collected per participant, preventing an analysis of intra-subject variability, which would have strengthened reliability conclusions.

Overall, the manuscript is significantly improved and provides valuable insights into the accuracy of Apple Watch VO₂ max estimates. The study is timely and well-conducted, contributing to the growing field of wearable health technology validation. However, to further refine the manuscript, the authors should explicitly emphasize the study’s limited generalizability and acknowledge the need for future research incorporating multiple Apple Watch estimates per participant. With these minor refinements, the manuscript is suitable for publication.

Reviewer #2: (No Response)

**Do you want your identity to be public for this peer review?** For information about this choice, including consent withdrawal, please see our Privacy Policy

Reviewer #1: No

Reviewer #2: **Yes: ** Jordan Brayanov

---

## [Author Response · Author response to Decision Letter 2]

11 Apr 2025

Response to Reviewers

Journal Requirements:

COMMENT: Please review your reference list to ensure that it is complete and correct. If you have cited papers that have been retracted, please include the rationale for doing so in the manuscript text, or remove these references and replace them with relevant current references. Any changes to the reference list should be mentioned in the rebuttal letter that accompanies your revised manuscript. If you need to cite a retracted article, indicate the article’s retracted status in the References list and also include a citation and full reference for the retraction notice.

RESPONSE: Each of the 52 references have now been cross-checked by a hand-search to ensure that they are correct. Alterations have been made to the following references:

Hill and Lupton (reference number 1), Grand View Research (reference number 25), Statista wearable device ownership (reference 24), Apple watchOS (reference 27), Apple Cardio Fitness estimates (reference 28), ACSM fitness trends (reference 23), Astrand protocol (reference 35).

No retracted papers have been cited.

----

Reviewer 1:

COMMENT: The revised manuscript presents a well-structured and thorough validation study on the accuracy of Apple Watch VO₂ max estimates compared to the gold-standard method of indirect calorimetry. The authors have effectively addressed the concerns raised in the initial review by improving methodological transparency, incorporating additional statistical analyses, and expanding the discussion to provide better context for their findings. The inclusion of a post-hoc power analysis strengthens the study’s credibility, confirming that the sample size of 28 participants was statistically sufficient to detect meaningful differences between Apple Watch and indirect calorimetry. Additionally, the authors conducted regression analyses to assess the influence of age and sex on measurement discrepancies, although no significant associations were found. These additions enhance the depth of the study and demonstrate a rigorous approach to validation.

A major improvement in the revised version is the expanded discussion, which now situates Apple Watch’s performance within the broader context of wearable technology and traditional submaximal exercise tests. The authors acknowledge the proprietary nature of Apple’s algorithm and how software updates may influence accuracy over time. Furthermore, they compare the observed error margins to those of conventional submaximal VO₂ max prediction methods, reinforcing the notion that Apple Watch may be more suitable for general fitness tracking than for clinical decision-making. The discussion also now includes a reference to the inherent variability of indirect calorimetry (~2.58 mL/kg/min), providing a useful benchmark for interpreting the Apple Watch’s underestimation of VO₂ max by 6.07 mL/kg/min.

RESPONSE: We are very grateful for your valuable comments during the previous round of review. Your review allowed us to greatly strengthen the manuscript. We conducted post-hoc power analysis, as well as regression analyses, and expanded the Discussion section. Additionally, the comparison of Apple Watch’s error margin to that of conventional testing adds substantially to the interpretation of our results — thank you for this recommendation.

COMMENT: Despite these strengths, some limitations remain. The sample is still skewed towards individuals with high cardiorespiratory fitness, which restricts the generalizability of findings to less fit or clinical populations. Although the authors acknowledge this in the discussion, future studies should aim for a more diverse participant pool. Another limitation is the uncontrolled nature of the Apple Watch data collection process, as participants generated VO₂ max estimates independently over several days without standardized exercise conditions. While this enhances ecological validity, it introduces external variables that may have influenced accuracy. Additionally, only one Apple Watch measurement was collected per participant, preventing an analysis of intra-subject variability, which would have strengthened reliability conclusions.

RESPONSE: We have implemented revisions to the discussion section of the manuscript in response to your comments regarding the composition of the sample, the Apple Watch data collection process, and the fact that one estimate was collected per participant. The limitations section of the discussion has been expanded to describe this with additional detail:

“However, the predominance of participants with high levels of cardiorespiratory fitness represents the study’s most significant limitation. Only one participant was classified as having ‘Fair’ cardiorespiratory fitness, while all others were categorised as ‘Good’, ‘Excellent’, or ‘Superior’. This limits the generalisability of our findings to populations with lower fitness levels, including clinical cohorts. Additionally, the sample primarily consisted of younger individuals, despite efforts to recruit participants across a broad age range. Another limitation is the uncontrolled procedure of generating Apple Watch VO2 max estimates. While this protocol reflects real-world use, it introduces unquantified external variables that may have influenced prediction accuracy. Lastly, only one VO2 max estimate was collected per participant, preventing analysis of intra-subject variability, which may have provided insights into reliability and changes in measurement accuracy over time.

[...] Future research should include individuals with diverse cardiorespiratory fitness levels to enhance generalisability, support the refinement of predictive algorithms, and develop frameworks for ongoing evaluation of wearable devices. Such efforts are essential to bridging the gap between technological innovation and evidence-based practice, ultimately enhancing the utility of wearable devices for individual health monitoring and public health applications.”

--

COMMENT: Overall, the manuscript is significantly improved and provides valuable insights into the accuracy of Apple Watch VO₂ max estimates. The study is timely and well-conducted, contributing to the growing field of wearable health technology validation. However, to further refine the manuscript, the authors should explicitly emphasize the study’s limited generalizability and acknowledge the need for future research incorporating multiple Apple Watch estimates per participant. With these minor refinements, the manuscript is suitable for publication.

RESPONSE: We do hope that we have effectively addressed your comments and suggestions. We endeavoured to emphasise the limited generalisability of our findings to individuals with low cardiorespiratory fitness, whilst highlighting limitations relating to the data collection, the absence of intra-subject variability analysis, and the need for future research to address these limitations. Thank you for your valuable contributions to this manuscript.

--

Reviewer 2

RESPONSE: The second reviewer indicated in Question 1 that all of their comments from the previous round of review were adequately addressed and that the manuscript is now acceptable for publication.

---

## [Decision Letter · Decision Letter 2]

15 Apr 2025

Investigating the accuracy of Apple Watch VO2 max measurements: A validation study

PONE-D-25-04283R2

Dear Dr. Lambe,

We’re pleased to inform you that your manuscript has been judged scientifically suitable for publication and will be formally accepted for publication once it meets all outstanding technical requirements.

Kind regards,

Emiliano Cè, Ph.D.

Academic Editor

PLOS ONE

Additional Editor Comments (optional):

Reviewers' comments:

Reviewer's Responses to Questions

**Comments to the Author**

Reviewer #1: All comments have been addressed

2. Is the manuscript technically sound, and do the data support the conclusions?

Reviewer #1: Yes

3. Has the statistical analysis been performed appropriately and rigorously?

Reviewer #1: N/A

4. Have the authors made all data underlying the findings in their manuscript fully available?

Reviewer #1: Yes

5. Is the manuscript presented in an intelligible fashion and written in standard English?

Reviewer #1: Yes

Reviewer #1: In the revised paper, the authors have covered all the comments made by the reviewer and therefore, in my opinion, the paper can be considered for publication.

**Do you want your identity to be public for this peer review?** For information about this choice, including consent withdrawal, please see our Privacy Policy

Reviewer #1: No

---

## [Editor Report · Acceptance letter]

PONE-D-25-04283R2

PLOS ONE

Dear Dr. Lambe,

I'm pleased to inform you that your manuscript has been deemed suitable for publication in PLOS ONE. Congratulations! Your manuscript is now being handed over to our production team.

Kind regards,

on behalf of

Prof. Emiliano Cè

Academic Editor

PLOS ONE